# Sedimentary Responses of Late Triassic Soft-Sedimentary Deformation to Paleoearthquake Events in the Southwestern North China Plate

**Wei Yu [1,2], Qingshao Liang [1,2,*], Jingchun Tian [1,2], Yonglin Han [3], Feng Wang [1,2] and Ming Zhao [3]**

[1] Institute of Sedimentary Geology, Chengdu University of Technology, Chengdu 610059, China
[2] State Key Laboratory of Oil and Gas Reservoir Geology and Exploitation, Chengdu University of Technology, Chengdu 610059, China
[3] Research Institute of Exploration and Development, PetroChina Changqing Oilfield Company, Xi'an 710021, China
* Correspondence: liangqs@cdut.edu.cn

**Abstract:** Tectonic events caused by paleoearthquakes are reflected in sediments. Outcrops and cores from the Chang-7 Member of the Late Triassic Yanchang Formation, Ordos Basin in Northern China, yield a wide variety of soft-sediment deformation structures (SSDSs), many of which are laterally extensive for more than 150 km. They include various types of folds, soft-sediment liquefaction flow deformation (liquefied sand dyke, liquefied breccia), gravity-driven deformation (load structures, ball-and-pillow structures), hydroplastic deformation (loop bedding, convolute deformation), and brittle deformation (intrastratal and stair-step faults, cracks). In most cases, deformation resulted in hybrid brittle-ductile structures exhibiting lateral variation in deformation style. These occur in delta front to semideep-to-deep lake sands and mudstones (shales). The seismites recognized in outcrops and cores indicate earthquakes with magnitudes (Ms) between 6 and 8, which are interpreted as a response to orogenic events related to the collision of the South China Block (SCB) and North China Block (NCB) during the Late Triassic period. Systematic study of the spatial and temporal distribution of these seismites improves the understanding of the tectonic context and evolutionary history of sedimentary basements. This study can provide a new perspective on the evolution of tectonic activities in the basin.

**Keywords:** seismic events; trigger mechanism; seismites; Chang-7 member; ordos basin

## 1. Introduction

Since Seilacher [1] proposed the concept of seismites, many studies have been devoted to this topic, and significant progress has ensued: (1) The identification criteria and vertical sequence of seismic rocks have been proposed [2–6]. (2) The genetic mechanism of sediment liquefaction deformation caused by seismic activity has been analyzed [6–20]. (3) Seismites in various rock types, such as claystone, coarse sandstone to fine sandstone, grainstone, shale, and evaporites, have been reported [21–23]. (4) The age of seismite formations ranges from the Meso-Neoproterozoic to modern [23]. (5) The development environments of seismites include those corresponding to lacustrine facies, fluvial facies, deep-sea basins, inland basins, beaches, foreshores, transition zones, continental shelves, and pelagic sedimentation [6,11,16,19,22,24–29]. (6) Recently, research on the structural characteristics, sequence, trigger mechanism, and scientific significance of SSDSs has strengthened [6,16,18–20,29–40].

The original definition of seismites has been extended from the deformation structure caused by seismic events to the deformation structure caused by seismically induced tsunamis, turbidity currents, and other events [2]. These deformation structures mainly include (1) deformation structures in the sedimentary layer that are directly formed by seismic vibration (such as annular layers and irregular convolute stratification), (2) the

overall transportation of sediments due to geological events caused by earthquakes (such as turbidity current events), and (3) sediment homogenization induced by seismic events. Seismites are sediments with SSDSs caused by seismic events or other events induced by earthquakes, which are typical representatives of sedimentary rocks generated during tectonic events [1,2].

Seismites are symbolic products of paleoearthquakes and their paleoenvironmental impacts. Seismites provide not only a reasonable dynamic explanation for the mechanism of tectonic evolution but also a scientific basis for restoring the strong activity of basin boundary faults on a smaller time scale [17,30,41,42].

The Ordos Basin evolved into an inland depression basin in the Mesozoic [43–45]. A large number of seismites were preserved in the Chang-7 Member of the Yanchang Formation in the Late Triassic, which is an excellent place to study the sedimentary response of paleoseismic events in a depression lake basin [43–49].

There are various triggering mechanisms of soft-sediment deformation. Earthquakes, volcanic activities, tides, storms, tsunamis, turbidity currents, etc., can form soft-sediment deformation [5,6]. Many times, these factors may be synchronous and related, such as volcanoes and earthquakes caused by plate collision, and earthquakes can lead to tsunamis, etc. In the study area, in the peak period of lake basin development in the Ordos Basin during the Chang-7 sedimentary period, the lake water depth was the largest, corresponding to a semideep-to-deep lake environment, which excludes the deformation of soft sediments triggered by waves and glaciers. The overlying and underlying normal sediment layers in the Chang-7 Member were not affected by deformation. This indicates that the deformation of soft sediments occurred in in-situ deposits that were not transported. Based on the above research, the mechanism for an allogenic trigger, such as earthquakes, can be considered [5,6,10,16,19,20,29,32,36,38,39].

Previous researchers have systematically studied the seismites in the Triassic Yanchang Formation in the Ordos Basin [46–50]. Tian et al. expounded on the superposition relationship between seismite sandstone and other genetic sand [20,51]. Although the sedimentology of the Yanchang Formation in the Ordos Basin has been studied extensively, only a few have recorded the presence of seismites [46–49]. The study of the formation mechanism, distribution range, and seismic grade of seismites in the Chang-7 Member remains weak.

The study of seismites in the Chang-7 Member is helpful in interpreting the tectonic activity history of the basin, and has important scientific value for the Triassic Ordos Basin tectonic evolution and basin–mountain coupling process [52–54]. We studied the deformation structures in lacustrine deposits. This study allowed us to (1) determine the distribution range of seismites in the study area; (2) deduce the most likely triggering mechanism; (3) interpret the seismic intensity at that time. Our study shows that understanding the deformation features of the Chang-7 Member and the physical conditions under which these features formed can enhance interpretations of the evolution of the Ordos Basin and enable the refinement of the paleoenvironmental setting and the paleotectonic history of the Ordos Basin.

## 2. Geological Setting

The formation and evolution of the Ordos Basin are controlled by the interaction of three dynamic systems: the Paleo-Asian Ocean, the Tethys Paleo-Pacific Ocean, and the Indian Pacific Ocean [55]. In the middle Late Triassic, the NCB was dominated by compressional tectonic deformation [55–57]. Due to the collision and orogeny of the Qinling orogenic belt, the southern part of the basin formed an edge of the northward thrust nappe fault [55–57]. At the same time, affected by the southwest Tethys tectonic domain, a series of strike-slip thrust faults formed [55–57]. The tectonic intensity diminished from south to north and was spatially heterogeneous. This indirectly controlled the Triassic sedimentary pattern and sequence filling characteristics in the Ordos Basin [56–59]. In the Middle Triassic, the Ordos Basin entered the period of inland differential subsidence

basin development [43,54]. The Indosinian movement opened a continental depression lake basin, which is also the most important structural development stage of the Ordos Basin. Under this structural background, the most developed oil generation system of the Mesozoic was formed [43,54].

The Ordos Basin is located in the western part of the NCB. It is a multicycle superimposed craton basin [43–45] (Figure 1a). This basin is characteristically large, with a gentle slope, shallow water depth, and multiple provenances [43–45]. The Yanchang Formation can be divided into ten subunits based on sedimentary cycles, Chang-10 to Chang-1, from bottom to top, which together record a complete cycle of lake development: initial formation and development stage (Chang-10 to Chang-8), peak stage (Chang-7 to Chang-4 + 5), and decline stage (Chang-3 to Chang-1) [44]. The study area is located in the southwestern Ordos Basin (Figure 1b,c). The Chang-7 Member can be subdivided into three submembers, named Chang-$7_3$, Chang-$7_2$, and Chang-$7_1$, from bottom to top [43–45].

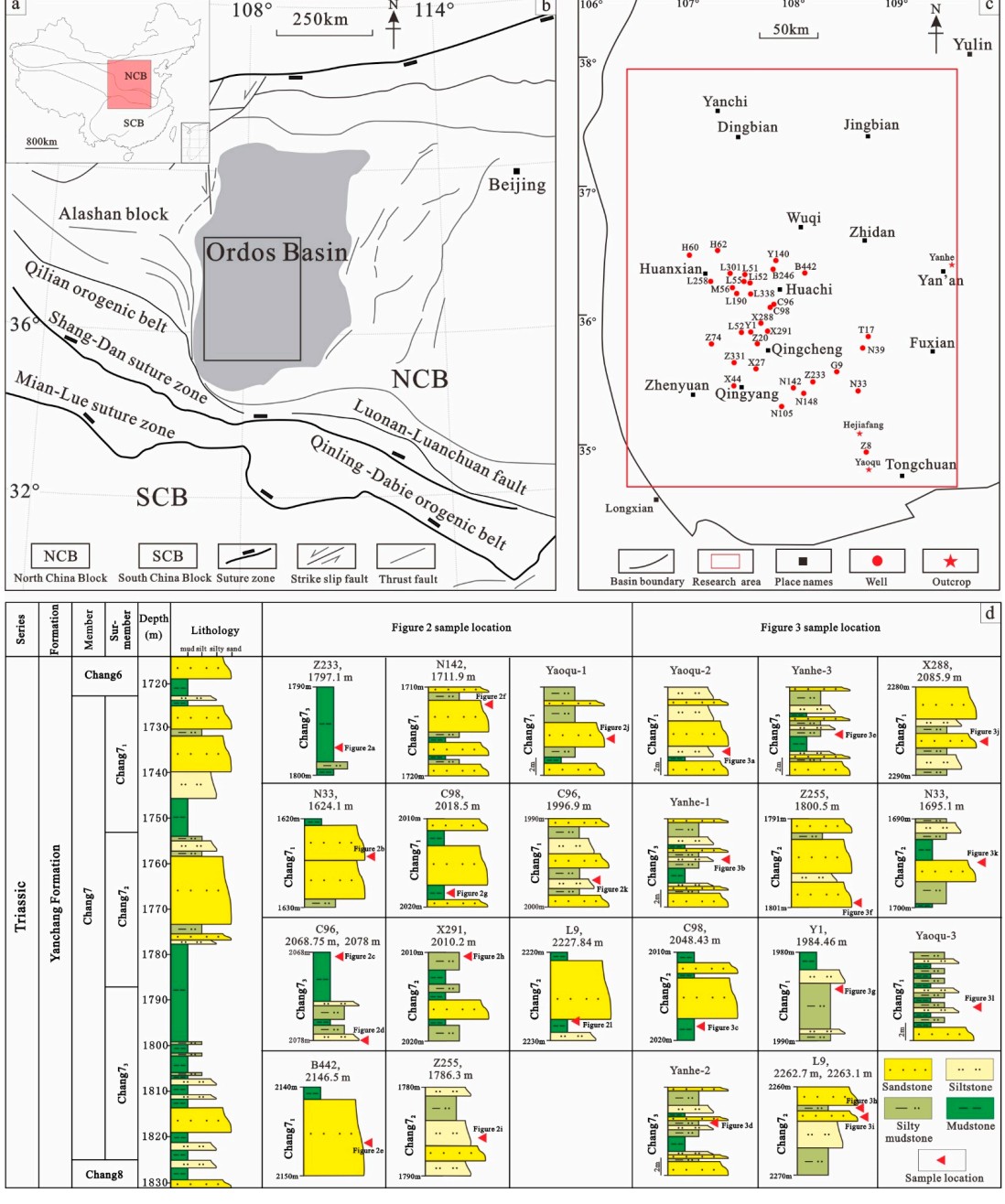

**Figure 1.** Geological setting (**a**,**b**), location (**c**), and sample location (**d**) of the study area.

## 3. Materials and Methods

For this study, 3 outcrops and the cored intervals of 97 exploration wells through the Chang-7 Member of the Late Triassic Yanchang Formation in the Ordos Basin were analyzed and measured in detail. These intervals were obtained from the PetroChina Changqing Oilfield Company of CNPC China, Ltd. SSDSs were observed in only 64 of the wells. The depth, thicknesses, lithological characteristics, and sedimentary structures were described and photographed, and the SSDSs were described in detail. All the data provide significant information to analyze the sedimentary characteristics of SSDSs and help deduce seismic and tectonic activity.

## 4. Types of SSDSs and Their Genesis

Outcrops and cores from the Chang-7 Member of the Late Triassic Yanchang Formation in the Ordos Basin yield a wide variety of SSDSs. These SDSS types can be divided into four categories according to their genetic properties: (1) soft-sediment liquefaction flow deformation, (2) gravity-driven deformation, (3) hydroplastic deformation, and (4) brittle deformation (Table 1).

**Table 1.** Genetic types and characteristics of seismites in the Chang-7 Member [27,60–62].

| Genetic Type | Deformation Structure Type | Characteristic |
|---|---|---|
| Soft sediment liquefaction flow deformation | Liquefied sand dyke | The sand layer in the soft sediment is liquefied and flows to the vein of the adjacent soft sediment layer, and its scale ranges from millimeter level to meter level. |
| | Liquefied breccia | The liquefied sand layer pierces the adjacent soft sedimentary layer to brecciate the soft sediment layer. |
| Gravity-driven deformation | Load structure | Soft sediment loses balance under the action of earthquakes, move and deform vertically, driven by gravity, and coarse-grained sediments form and sink. |
| | Ball-and-pillow structure | The load body separates from the parent rock to form sand balls and sand pillows. |
| Hydroplastic deformation | Loop bedding | Under the action of seismic shear stress, hydroplastic sediments are formed by tensile stress. |
| | Convolute deformation | Under the action of seismic shear stress, the soft sedimentary layer is formed by plastic sliding. |
| Brittle deformation | Intrastratal fault and stair-step faults | Under the compression force caused by the earthquake, the dislocation caused by the earthquake in the consolidated rock and the sedimentary layer at the top of the stratum that has not been fully consolidated. |
| | Cracks | A series of vertical cracks with irregular fracture surfaces are produced by earthquakes in the consolidated stratum, mainly in the form of tensile cracks. |

### 4.1. Soft-Sediment Liquefaction Fow Deformation

The pores of saturated soft sediments are filled with water. When subjected to strong earthquake vibration, especially shear force, the originally stable pore water pressure will rise. The rising pressure creates excess pore water pressure, resulting in the liquefaction and flow of soft sediments [35,62,63].

The conditions for liquefaction of soft sediments are described as follows [62,64,65]: (1) The average diameter of liquefiable particles is between 0.05 and 1 mm, (2) the content of clay particles is less than 10%, (3) the inhomogeneous particle coefficient is less than 10, (4) the sand layers have high porosity, (5) the maximum liquefaction depth is 20 m from the land surface, the maximum groundwater depth is approximately 5 m, and (6) an earthquake magnitude of 5 or greater.

Soft-sediment liquefaction deformation in the Chang-7 Member mainly produced liquefied sand dyke and liquefied breccia.

(1)  Liquefied sand dyke

Several liquefied sand dykes were found in the cores of the Chang-7 Member, including in those of Well N33, Well Z233, Well B442, and Well C96. The observed sand dykes exceed 15 cm in vertical extent and are up to 2 cm wide (Figures 1d and 2a–d).

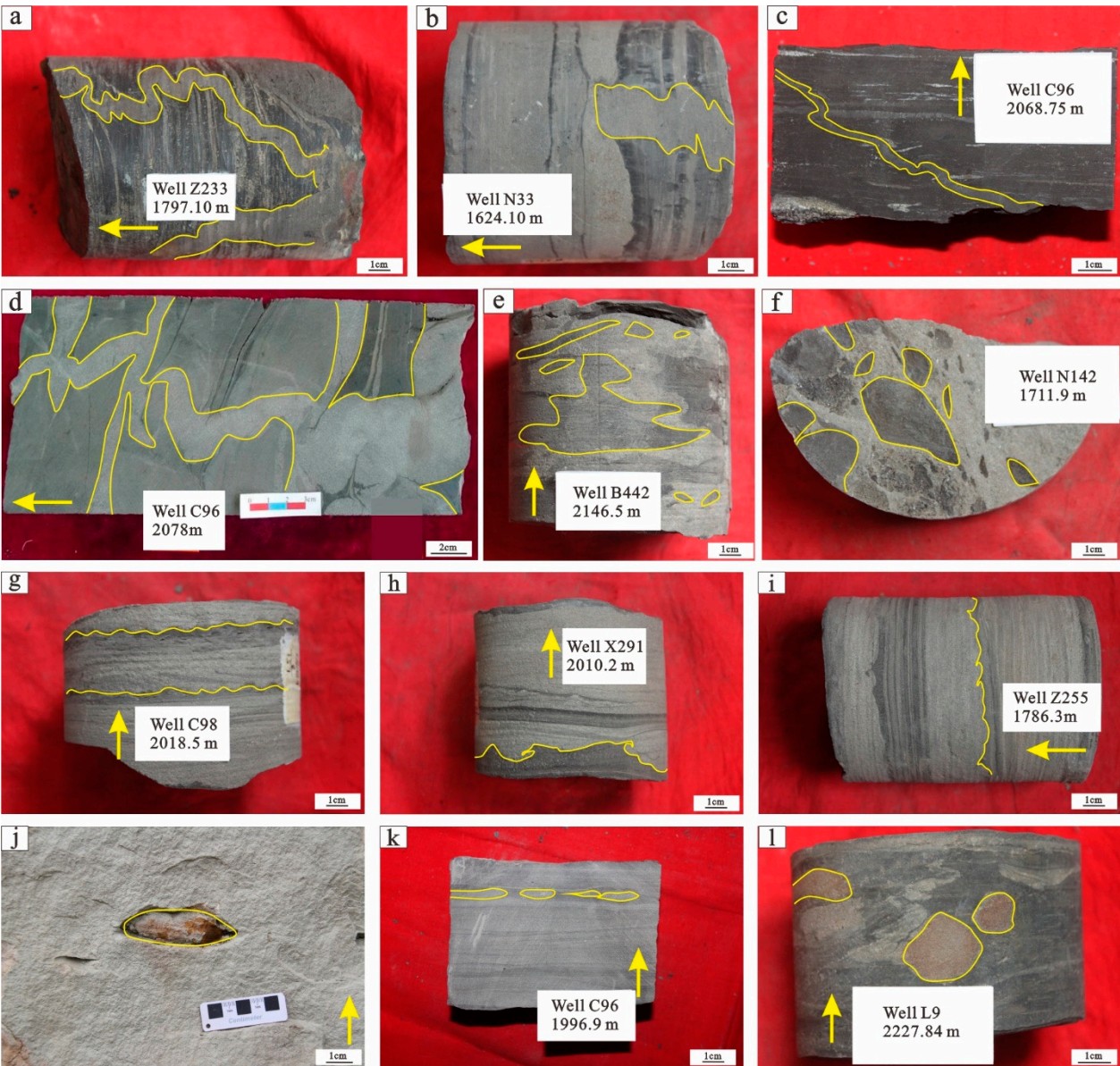

**Figure 2.** Soft-sedimentary deformation structures (**a**): Well Z233, Chang-7 Member, 1797.10 m, liquefied sand dyke; (**b**): Well N33, Chang-7 Member, 1624.10 m, liquefied sand dyke; (**c**): Well C96, Chang-7 Member, 2068.75m, liquefied sand dyke; (**d**): Well C96, Chang-7 Member, 2078.00 m, liquefied sand dyke; (**e**): Well B442, Chang-7 Member, 2146.50 m, liquefied sand dyke; (**f**): N142, Chang-7 Member, 1711.90 m, liquefied breccia; (**g**): Well C98, Chang-7 Member, 2018.50 m, load structure; (**h**): Well X291, Chang-7 Member, 2010.20 m, injection structure; (**i**): Well Z255, Chang-7 Member, 1786.30 m, flame structure; (**j**): Yaoqu-1, Chang-7 Member, ball structure; (**k**): Well C96, Chang-7 Member, 1996.90 m, pillow layer; (**l**): Well L9, Chang-7 Member, 2227.84 m, ball structure.

Seismic action changes the original stable state of sediments in strata, weakens the stress between sediment particles, transfers the stress between sandstone frameworks, and

forms excess pore water pressure. The uneven distribution of overpressure leads to rock fracture. Liquefied sandstone moves toward rock fractures or lower pressures under the action of stratigraphic pressure, forming liquefied sand dyke [60–62,66].

(2) Liquefied breccia

The liquefied breccia is argillaceous and formed in situ, without transportation, and the cement of the liquefied breccia in the studied rock formed from liquefied sand. Liquefied breccias can independently occur as semiplastic mud breccias in situ or appear in association with other ductile deformation structures, such as load structures and liquefied sand dyke (Figures 1d and 2e,f).

Liquefied breccia is a common seismite. The reason for brecciation is that the liquefied sand layer pierces the adjacent soft sediment layers and brecciates the soft sediment layer [62,66,67]. Brecciation of the argillaceous layer or the soft sedimentary layer indicates that the argillaceous layer or the soft sediment layer was overpressured to reach the state of fluidization during deformation and damaged harder silty and sandy units. The formation of these breccias was not associated with slope collapse, fault activity, or erosion.

### 4.2. Gravity-Driven Deformation

The soft sediments can lose balance under the action of earthquakes; then, they move and deform vertically under the action of gravity and form a load structure. When the load cast continues to sink and break away from the mother rock layers, ball-and-pillow structures form [60–62,68,69]. In the cores of the Chang-7 Member, the seismite types caused by gravity are mainly load structures and ball-and-pillow structures.

(1) Load structure

Due to the abnormal compression of the load structure, the overlying sand layer is subjected to tensile stress, and the argillaceous sediment penetrates the sand layer upward to form a flame structure (Figures 1d and 2g–i). A flame structure can be skewed under the action of horizontal shear force, and the direction of the flame points to the downstream direction of the slope [60,66] (Figure 2i). The static pressure of the unconsolidated sediment layer is destroyed due to shaking or gravity differentiation, and the higher-density sediments sink into the underlying lower-density sediments to form a load structure [60,66].

(2) Ball-and-pillow structure

The Chang-7 Member ball-and-pillow structure has a thickness of approximately 2 to 6 cm and a width of approximately 4.3 to 8.6 cm (note: the load structure with a width greater than 1 cm is of seismic origin [70] (Figures 1d and 2j–l).

The ball-and-pillow structure produced by a seismic event is different from the ball-and-pillow structure produced by an allochthonous differentiation [60,62,66]. Under the joint action of density gradation, gravity and seismic vibration, the liquefaction and fluidization of argillaceous sediments in the underlying layer lead to the loss of support of overlying sandy sediments to form a ball pillow structure, which is a seismic liquefaction deformation structure induced by strong seismic shear force [62,66]. The diameters of the ball-and-pillow structures of the studied seismites are between a few millimeters and a few centimeters. Under the influence of earthquakes, these sediments show various irregular deformation forms, such as drag, elongation, fragmentation, and tumors [66].

### 4.3. Hydroplastic Deformation

Hydroplastic deformation occurs in water-saturated soft sediments. The seismic shear stress increases the pore fluid pressure and weakens the support strength between soft sediment particles, but before reaching the degree of complete liquefaction and flow, continuous deformation without fracturing occurs under the action of the regional stress [62,66]. Hydroplastic deformation mainly occurs in situ. The conditions causing hydroplastic deformation are (1) deformation in surrounding soft sediments dragged by liquefaction

and thixotropy, (2) sustained and instantaneous stress action, and (3) low-angle deformation [60–62,66]. The hydroplastic deformation types of the Chang-7 Member are mainly loop bedding and convolute deformation.

(1)    Loop bedding

Loop bedding is a ductile deformation structure, which presents loop or chain shapes [10]. Widely distributed loop bedding is found in the upper part of the Zhangjiatan shale in the Chang-7 Member of the Yanhe section (Figures 1d and 3a–c), which shows that the Chang-7 Member in the northeastern part of the lake basin was less affected by earthquakes.

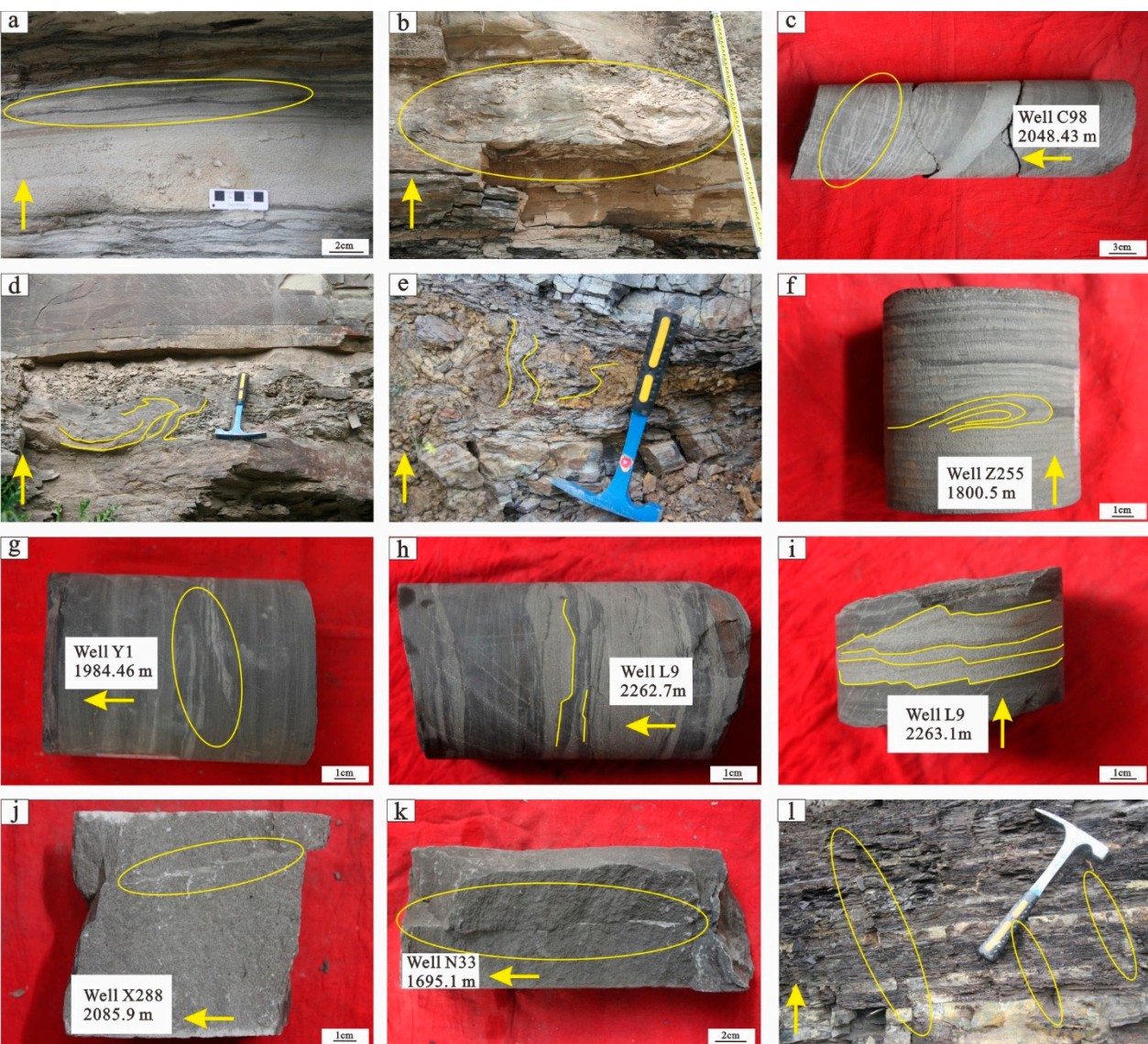

**Figure 3.** Soft-sedimentary deformation structures (**a**): Yaoqu-2, Chang-7 Member, loop bedding; (**b**): Yanhe-1, Chang-7 Member, loop bedding; (**c**): Well C98, Chang-7 Member, 2048.43 m, loop bedding; (**d**): Yanhe-2, Chang-7 Member, laminar sandstone convolute deformation; (**e**): Yanhe-3, Chang-7 Member, laminar sandstone convolute deformation; (**f**): Well Z255, Chang-7 Member, 1800.50 m, recumbent fold; (**g**): Well Y1, Chang-7 Member, 1984.46 m, intrastratal fault. (**h**): Well L9, Chang-7 Member, 2262.70 m, stair-step faults. (**i**): Well L9, Chang-7 Member, 2263.10 m, stair-step faults. (**j**): Well X288, Chang-7 Member, 2085.90 m, cracks. (**k**): N33, Chang-7 Member, 1695.10 m, cracks. (**l**): Yaoqu-3, Chang-7 Member, cracks.

Loop bedding is unconsolidated fine-grained thin interbedded sandy argillaceous sediment formed by the tensile stress induced by weak earthquakes. These structures consist of bundles of laminae that exhibit local constriction, forming a flattened elliptical geometry, which is 5–50 cm wide and 0.5–30 cm tall in this study area. It is a seismic deformation structure formed in a hydrostatic semideep-to-deep lake environment [51,60,61].

(2)　Convolute deformation

Such deformations are widely observed in the Chang-7 Member core and outcrop in the Ordos Basin. Liquefaction convolute deformation (Figures 1d and 3d) and wavy laminae (Figures 1d and 3e) are found in the Yanhe section, and recumbent folds are found in the core (Figures 1d and 3f).

When sediment layers are not consolidated, convolute deformations form under compressive stress and display plastic flow folds. Convolute deformation occurs in siltstone or silty mudstone, which are composed of laminar or thin soft-sediment deformation layers interbedded with undeformed layers. The shape of convolute deformation is generally curved, curled, undulating laminae, or recumbent folds [60–62,66].

### 4.4. Brittle Deformation

Brittle deformation results in faults and joints that are triggered by seismic activity, and it occurs in semiconsolidated or soft sediments at the top of the sedimentary sequence [6,60,61]. Intrastratal faults, stair-step faults, and cracks are typical types of brittle deformation in the Chang-7 Member. Brittle deformations represent earthquakes with magnitudes greater than 6 [1].

(1)　Intrastratal fault and stair-step fault

In the studied rock, a single fault is often shown as an intrastratal fault, the fracture distance is generally 2–10 mm, and the dip angle is low. Stair-step microfaults are a series of near-parallel faults; here, the fracture distance is generally 2–5 mm, the dip angle is low, the upper plate is a descending positive fault, and the profile is parallel to the stepped arrangement [1,60–62,66]. Intrastratal faults and stair-step microfaults have been found in the core and field outcrops of the Chang-7 Member. The spacing of intrastratal faults is generally greater than 1 cm [1] (Figures 1d and 3g). Stair-step microfaults are distributed in sand-mud interlayers, and the fault distance is generally less than 1 cm. Many stair-step microfaults are arranged in parallel. The length of each microfault is approximately 5 mm; they are dense and steeply dipping, and it is a normal fault [1] (Figures 1d and 3h,i).

Intrastratal faults and stair-step faults indicate brittle failure of stiff, perhaps even indurated sediments as a result of elevated stresses [1,62]. Intrastratal faults and stair-step faults are formed in the process of sedimentary stratigraphic vibration, mainly in the form of tensional fractures, which can be developed individually or arranged in parallel as stair-step faults, which are limited to intrastratigraphic development, without cutting through the upper and lower rock layers [1,60–62,66].

(2)　Cracks

Microcracks are typical sedimentary structures associated with palaeoseismic events. Microcracks represent earthquakes with magnitudes greater than 7 [60–62,66,71]. Cracks related to seismic events are found in sandstone and mudstone of the Chang-7 Member in the Yaoqu section, Well X288, and Well N33 (Figures 1d and 3j–l). They are perpendicular to layers, with varying widths from a few to tens of millimeters. Microfractures are formed by the tensile stress and liquefaction of sediments during earthquakes. The cracks are "V" shaped, which can indicate the top and bottom of the stratum [60–62,66].

## 5. Discussion

### 5.1. Earthquake Triggering Mechanism

Ancient earthquakes occurred against the tectonic background of collisional orogeny, and the strong episodic activity was the direct cause of earthquakes [54,56,58,72]. The

Indosinian movement was the most important tectonic movement in the Middle Triassic and Late Triassic in South China. During this period, the ancient Pacific Ocean basin began to subduct below the Asian continent [54,56,58]. During the Indosinian period, the PaleoTethys Ocean was subducted and closed, and the SCB and NCB collided along the Mian-Lue suture zone, resulting in orogenesis and connecting the South China Craton with the North China Craton [54,56,58]. In the middle and Late Triassic, the South China and North China plates were combined, the tectonic deformation and magmatic activity in the Qinling collision orogenic belt were strong, and earthquakes occurred frequently; thus, the Ordos Basin adjacent to the area north of the orogenic belt formed a steeply sloped terrain and semideep-to-deep lake area on the southwest edge, where the records of past deformation and sedimentation were simultaneously retained [54,57,72]. In this process, the earthquakes caused by stress adjustment release formed a large number of seismites [54].

The formation of seismites was the response of the basin–mountain coupling process. The intensity and cyclicity of the Qinling collisional orogeny not only indirectly controlled the sedimentary pattern and sequence filling characteristics in this period but also played a special role in the formation of seismites in the basin [54,57]. The tuff at the bottom of the Chang-7 Member is evidence of volcanic activity, and volcanic activity will lead to earthquakes. The zircon age of the tuff at the bottom of the Chang-7 Member is 239.3~243 ± 1.3 Ma, corresponding to the Middle Triassic Latin stage, and the tectonic activity corresponds to episode I of the Indosinian movement [36,57]. The orogenic and volcanic activity of Qinling Indosinian episode I are the main triggering mechanisms of seismic events in the Chang-7 Member (Figure 4).

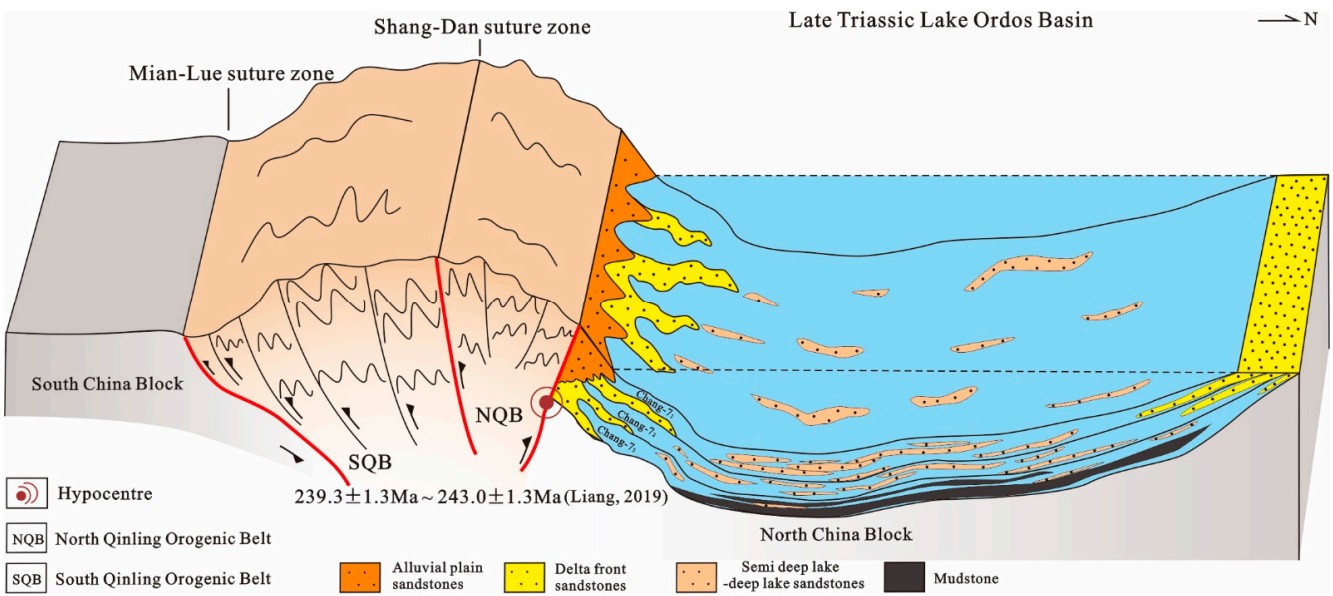

**Figure 4.** Model of the collisional setting during accumulation of the Chang-7 Member [36].

*5.2. Relationship between the Distribution of Seismites and Seismic Intensity*

The type of seismites is related to the seismic intensity. Some scholars have proposed using sedimentary structures to estimate seismic intensity [10]. Microcracks and stair-step microfaults occur in superstrong seismic events greater than Ms8. A ball-and-pillow structure is generated by medium earthquakes of Ms 6–7. Seismic events of Ms 5–6 will form liquefied convolute deformation and liquefied sand dyke in the sedimentary layer, while small seismic events that are less than Ms 5 will form loop bedding and convolute deformation [10,71].

Kuribayashi and Tatsuoka [73] found that there is a clear corresponding relationship between the maximum epicentral distance (R) of liquefaction deformation and seismic grade (M), and the R-M relationship diagram is created. According to the R-M diagram, the

maximum epicentral distance of liquefaction deformation is 50 km, and the corresponding earthquake magnitude grade is Ms 7. At 200 km, the earthquake magnitude is Ms 8 [74,75].

The maximum epicentral distance of liquefaction deformation in the Chang-7 Member of the Ordos Basin exceeds 100 km, so the maximum earthquake level exceeds Ms 7 (Figures 5 and 6).

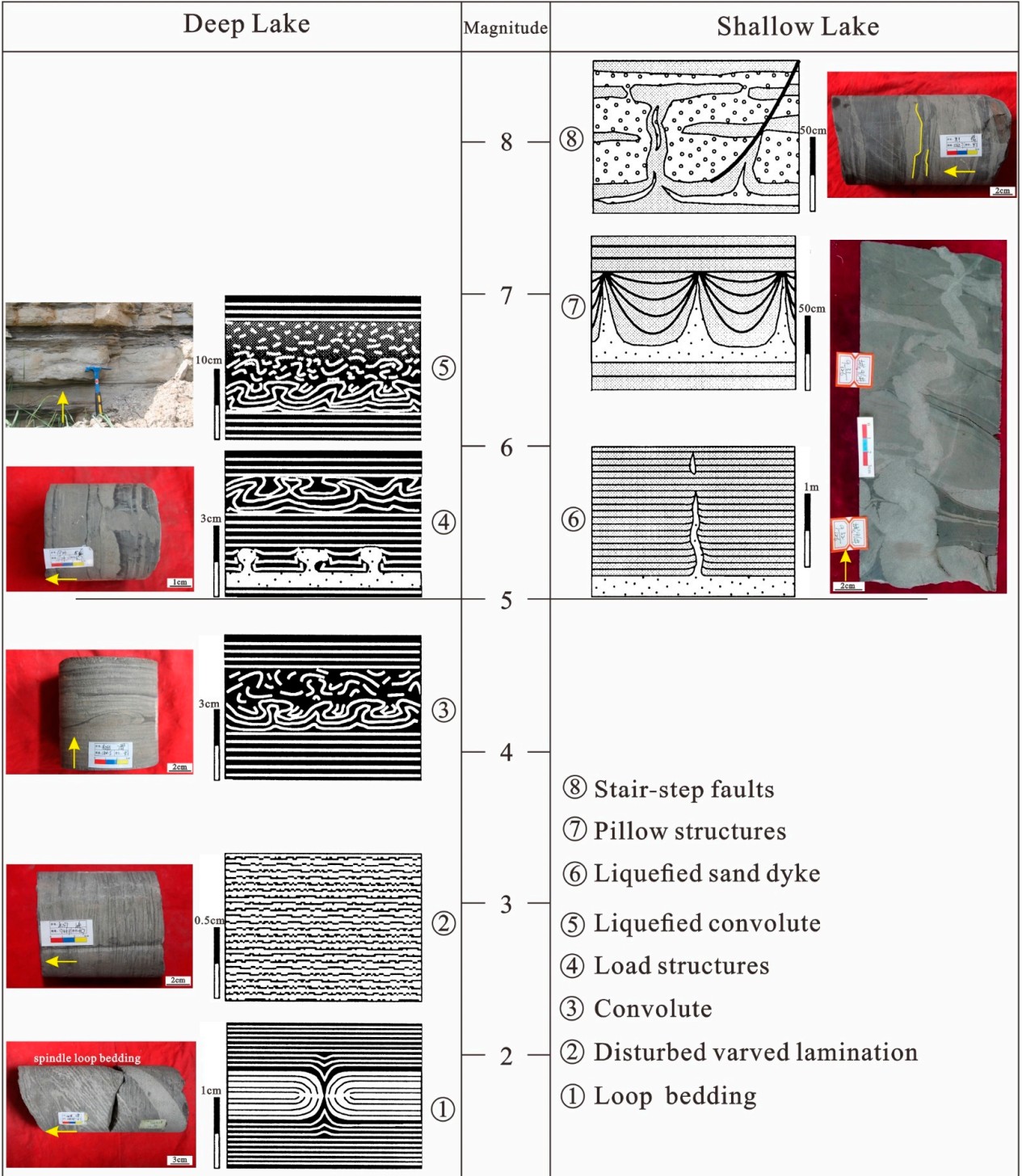

**Figure 5.** Relationship between the main types of seismites and seismic intensity in the Chang-7 Member.

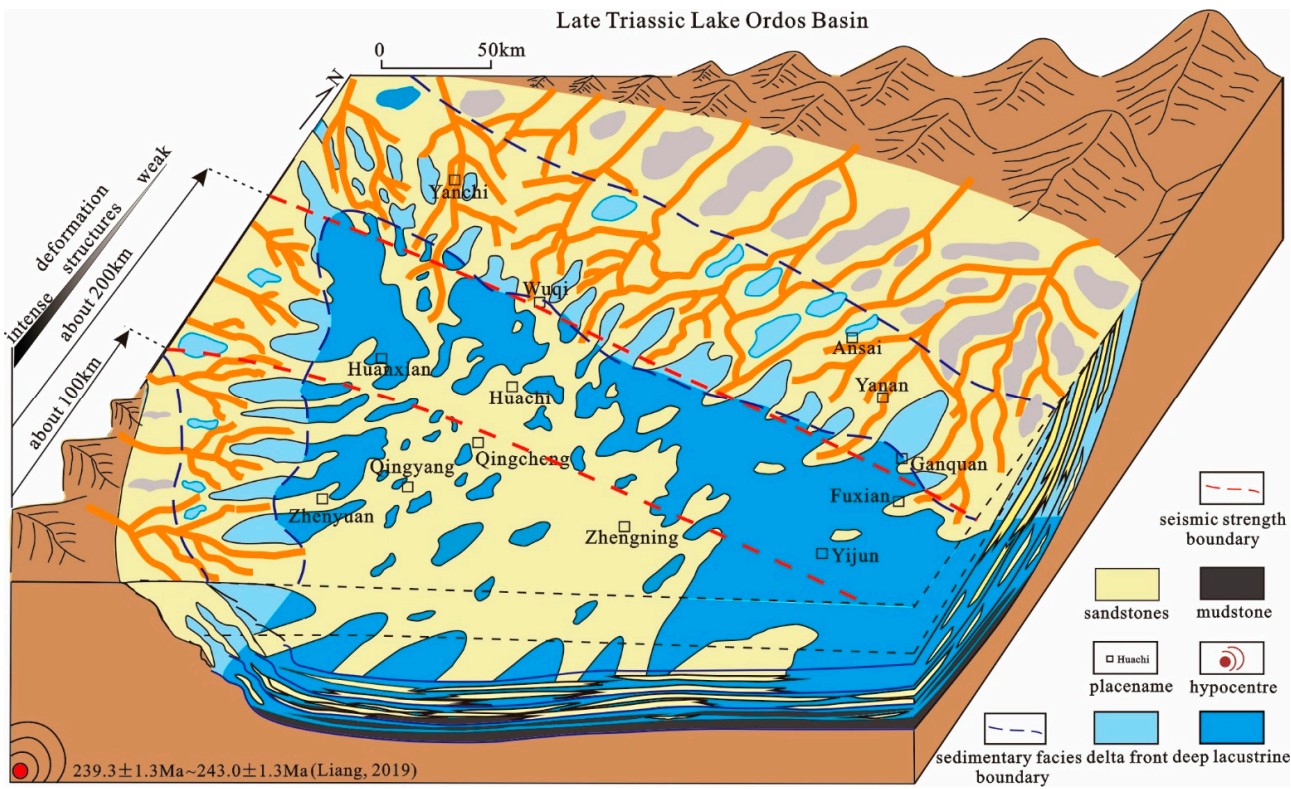

**Figure 6.** Plane distribution of seismic intensity in the Chang-7 Member [36,45].

According to the core distribution of seismites in the Chang-7 Member, different types of seismicity-induced SSDSs developed in semideep-to-deep lake areas due to different seismic intensities. Loop bedding, disturbed varved lamination, and convolute deformation developed in areas with weak seismic intensity. Load structures and liquefaction convolute deformation developed in areas with strong seismic intensity. In the semideep-to-deep lake area, liquefied sand dyke, pillow structures, and stair-step microfaults mainly developed (Table 2). These deformation types of seismites match the sequence of seismic intensity maps proposed by Rodrguez-Pascua et al. [10].

**Table 2.** Development location and type of seismicity-induced SSDSs in the Chang-7 Member [36].

| Well | Stratum | Depth (m) | R (km) | Sedimentary Environment | Deformation Structure Type |
|---|---|---|---|---|---|
| B246 | Chang-$7_3$ | 2238.6 | >175 | Semideep-to-deep lake | Liquefied sand dyke |
| H60 | Chang-$7_2$ | 2782.5 | >183 | Delta front | Liquefied sand dyke |
| H62 | Chang-$7_3$ | 2594.31 | >188 | Semideep-to-deep lake | Stair-microfaults |
| L51 | Chang-$7_1$ | 2267.96 | >173 | Semideep-to-deep lake | Cracks |
| Li52 | Chang-$7_2$ | 2294.03 | >137 | Semideep-to-deep lake | Stair-step faults |
| Li52 | Chang-$7_2$ | 2294.13 | >137 | Semideep-to-deep lake | Convolute deformation |
| Li52 | Chang-$7_2$ | 2294.26 | >137 | Semideep-to-deep lake | Convolute deformation |
| L52 | Chang-$7_3$ | 2309.1 | >165 | Semideep-to-deep lake | Cracks |
| Z20 | Chang-$7_1$ | 1728.3 | >64 | Semideep-to-deep lake | Intrastratal fault |
| Y140 | Chang-$7_1$ | 2211.4 | >107 | Semideep lake | Load structure and ball-and-pillow structure |
| B246 | Chang-$7_3$ | 2224.2 | >187 | Semideep-to-deep lake | Liquefied sand dyke |
| X44 | Chang-$7_1$ | 1973.5 | >175 | Semideep-to-deep lake | Liquefied sand dyke |
| Z74 | Chang-$7_2$ | 2523.5 | >80 | Semideep-to-deep lake | Load structure and ball-and-pillow structure |
| G9 | Chang-7 | 1767.2 | >125 | Semideep-to-deep lake | Intrastratal fault |
| Z1 | Chang-7 | 1256.2 | >76 | Semideep-to-deep lake | Liquefied sand dyke |
| Z8 | Chang-$7_1$ | 1223.5 | >39 | Semideep-to-deep lake | Liquefied breccia |
| T17 | Chang-$7_1$ | 1421.8 | >40 | Semideep-to-deep lake | Ball-and-pillow structure |

**Table 2.** *Cont.*

| Well | Stratum | Depth (m) | R (km) | Sedimentary Environment | Deformation Structure Type |
|---|---|---|---|---|---|
| Z331 | Chang-$7_1$ | 2155.2 | >116 | Semideep-to-deep lake | Stair-microfaults |
| M56 | Chang-$7_2$ | 2284.6 | >98 | Semideep-to-deep lake | Stair-step faults |
| N105 | Chang-$7_3$ | 1528.3 | >157 | Semideep-to-deep lake | Liquefied sand dyke |
| N33 | Chang-$7_3$ | 1724.1 | >62 | Semideep-to-deep lake | Stair-step faults |
| N39 | Chang-$7_3$ | 1667.3 | >68 | Semideep-to-deep lake | Stair-step faults |
| X27 | Chang-$7_2$ | 2044.4 | >92 | Semideep-to-deep lake | Loop bedding |
| N33 | Chang-$7_1$ | 1624.1 | >116 | Semideep-to-deep lake | Liquefied sand dyke |
| Z233 | Chang-$7_3$ | 1797.1 | >68 | Semideep-to-deep lake | Liquefied sand dyke |
| N148 | Chang-$7_1$ | 1635.8 | >83 | Semideep-to-deep lake | Liquefied sand dyke |
| C96 | Chang-$7_3$ | 2068.75 | >74 | Semideep-to-deep lake | Liquefied sand dyke |
| C96 | Chang-$7_3$ | 2080.6 | >147 | Semideep-to-deep lake | Liquefied sand dyke |
| C96 | Chang-$7_3$ | 2079 | >147 | Semideep-to-deep lake | Liquefied sand dyke |
| C96 | Chang-$7_3$ | 2067 | >147 | Semideep-to-deep lake | Convolute deformation |
| L190 | Chang-$7_1$ | 2246.6 | >147 | Semideep-to-deep lake | Liquefied breccia |
| C98 | Chang-$7_1$ | 2036.6 | >156 | Semideep-to-deep lake | Liquefied breccia |
| B442 | Chang-$7_1$ | 2146.5 | >144 | Semideep-to-deep lake | Liquefied breccia |
| N142 | Chang-$7_3$ | 1711.9 | >169 | Semideep-to-deep lake | Liquefied breccia |
| X291 | Chang-$7_2$ | 2010.2 | >78 | Semideep-to-deep lake | Flame structure |
| N148 | Chang-$7_2$ | 1719.12 | >124 | Semideep-to-deep lake | Load structure |
| L301 | Chang-$7_1$ | 2355.1 | >74 | Semideep-to-deep lake | Ball-and-pillow structure |
| L258 | Chang-$7_1$ | 2346.6 | >168 | Semideep-to-deep lake | Convolute deformation |
| L338 | Chang-$7_2$ | 2330.9 | >160 | Semideep-to-deep lake | Cracks |
| X288 | Chang-$7_1$ | 2085.9 | >155 | Semideep-to-deep lake | Cracks |
| Y1 | Chang-$7_1$ | 2000.35 | >131 | Semideep-to-deep lake | Stair-step faults |
| Z20 | Chang-$7_2$ | 1728.3 | >120 | Semideep-to-deep lake | Stair-step faults |

(R = vertical distance from the well to the bottom of red rectangle study area in Figure 1).

There is a close relationship between the magnitude of an earthquake and the extent of the distribution range of the related liquefaction deformation. The higher the earthquake magnitude is, the larger the distribution range of liquefaction deformation. According to the relevant research of Rodriguez-Pascua et al., when the earthquake magnitude is Ms 8, the distribution range of liquefaction deformation can reach 200 km [10]. The seismite distribution characteristics were analyzed by observations of the Chang-7 Member cores and outcrops [47,51–53] (Table 2).

The deposits in which seismites are found in the study area were formed in the environments of delta front and semideep-to-deep lake facies (Table 2). Seismites mainly developed in the semideep-to-deep lake environment, and they are sporadically observed in the delta front environment (Table 2). Seismites are mainly distributed in the southwestern area of the basin, and there are few seismites in the northeastern area; therefore, we can infer the development range of seismites in the Chang-7 Member (Figure 6). According to the relationship between the characteristics of seismites and seismic intensity [73], the influence range of seismic intensity is the strongest in the southwestern area and gradually weakens in the northeastern area of the basin. The earthquake occurred in the southwestern part of the basin.

*5.3. Relationship between SSDSs and Sedimentary Environment*

In addition to external causes, soft-sediment deformation is also closely related to the development characteristics of rock types and sedimentary facies. In the study area, soft-sediment liquefaction flow deformation is widely developed in the delta front sediments of the Chang-7 Member. The lithology of the delta front is mainly sandstone and siltstone. The reason for deformation is that the pore fluid in the rock cannot support the weight of a single grain [6]. In the semideep lake area, the lithology is mainly sandstone and siltstone, and a small amount of mudstone developed at the same time. Under the joint action of density gradation, gravity and seismic vibration, the overlying sandy sediment deformed because

the underlying muddy sediment could not be supported. In the semideep-to-deep lake area, in addition to fine sandstone and siltstone, a large number of mudstones were deposited. In the mudstones, the particles are closely arranged, causing the semideep-to-deep lake area to develop not only soft-sediment deformation, but also brittle deformation such as faults and fractures. At the same time, in the deep lake area with sand-mud interbedding, because it was far from the source, the earthquake intensity was small, and hydroplastic deformation occurred.

## 6. Conclusions

The sedimentary response characteristics of seismites can be divided into soft-sediment liquefaction flow deformation, gravity-driven deformation, hydroplastic deformation, and brittle deformation. Soft-sediment liquefaction flow deformation includes liquefied sand dyke and liquefied breccia. The characteristics of seismites caused by gravity mainly include load structures and sand ball-and-pillow structures. The hydroplastic deformation type mainly includes loop bedding and convolute deformation. The brittle deformation types are mainly intrastratal faults, stair-step microfaults, and microcracks.

Indosinian episode I of the Qinling regional orogeny was the triggering mechanism of earthquake events relevant to this study. The Chang-7 Member seismites in the Ordos Basin mainly developed in the semideep-to-deep lake area. The earthquakes in the southwestern part of the basin were the most active, with the highest earthquake intensity of more than Ms 7, and its influence range gradually weakened from the southwest to the northeast.

**Author Contributions:** W.Y.: Conceptualization, formal analysis, investigation, writing—original draft. Q.L.: Supervision, conceptualization, funding acquisition, writing—review and editing. J.T.: Supervision, conceptualization, funding acquisition, writing—review and editing. F.W.: Conceptualization, writing—review and editing. Y.H.: Resources. M.Z.: Resources. All authors have read and agreed to the published version of the manuscript.

**Funding:** This study was supported by the Ministry of Science and Technology (China) (No. 2016ZX0504605-001).

**Data Availability Statement:** All data generated or analyzed during this study are included in this published article.

**Acknowledgments:** Authors thank the Editors and anonymous referees for their helpful comments. Thanks to the approval of core samples from the Xifeng core library of PetroChina Changqing Oilfield Company.

**Conflicts of Interest:** The authors declare that they have no known competing financial interests or personal relationships that could have appeared to influence the work reported in this paper.

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
