# Peer review of "Sedimentary Responses of Late Triassic Soft-Sedimentary Deformation to Paleoearthquake Events in the Southwestern North China Plate"

_minerals, doi:10.3390/min12081044_

Round 1
Reviewer 1 Report
Dear Authors,
I read the manuscript with interest. The paper presents examples of soft-sediment deformation structures of different characteristics interpreted from cores. The description of the study area is good. However, some points should be much better presented and clarified. Moreover, the main interpretation of the seismicity-related origin of the described SSDSs should be better discussed and supported by evidence.
Below I attach my major comments:
1. It is very hard to distinguish what is your work and which part comes from the cited papers. You call the paper 'review' paper, but it rather seems to be 'original article'.
2. Sections 4 and 5:
Critical discussion of all possible mechanisms of soft-sediment deformation is missing. The 'seismite identification criteria' such as lateral extent of layers with deformations; distribution of the single SSDSs within the deformed layers; repetition of layers with SSDSs in vertical section seems to be missing.
As a reader, I do not feel convinced in any way that the described SSDSs are seismicity-induced. You should prove it delivering evidence. Without such discussion your interpretations seem more like a hypothesis.
The conclusions drawn about the EQ magnitude and location ot the hypocenter are very advanced, but based on a brief and insufficient discussion of all possible deformation mechanisms. I would expect some more discussion that supports your interpretation.
4. Editorial changes that I recommend:
- Some complete sentences and/or some single expressions are simply incomprehensible. It is necessary to re-read and linguistic proofread the entire manuscript.
- Orientation of samples (indication where is top, e.g. arrow symbol) is missing at Figs. 2 and 3.
I also add the pdf file with all detailed comments and sugestions of changes that - as I believe - can improve your manuscript.

Reviewer 2 Report
This manuscript takes SSDS (soft sedimentary deformation structures) of Chang-7 in Ordos Basin as the research object. The genesis of SSDS is discussed by describing the characteristics of SSDS, and the magnitude of earthquakes in sedimentary period is judged based on the distribution of SSDS.
In general, this manuscript is well written, with sufficient evidence and detailed discussions. I will recommend it after the following concerns are addressed.
(1) In the Introduction section, please do not cite such a large number of references in a single sentence; some of them are quite old (Line 39-42,47-50,52-55). Instead, some latest finds in this study area should be included.
Xiong, Y., Tan, X., Zuo, Z., et al. (2019). Middle Ordovician multi-stage penecontemporaneous karstification in North China: Implications for reservoir genesis and sea level fluctuations. Journal of Asian earth sciences, 183, 103969.
Xiong, Y., Tan, X., Zhong, S., et al. (2022). Dynamic paleokarst geochemistry within 130 Myr in the Middle Ordovician Shanganning carbonate platform, North China. Palaeogeography, Palaeoclimatology, Palaeoecology, 591, 110879.
(2) It is very good to mark the position of each picture in Figure 1d. It is suggested to mark the corresponding picture number behind the red triangle, which will be clearer.
(3) Lines 108-109 “The Yanchang Formation can be divided into ten subunits based on sedimentary cycles, Chang-10 to Chang-1, which …”. Please explain that it is a bottom-up relationship.
(4) You are suggested to rewrite the summary of Chapter 4 (Lines 126 to 129) to make a statement that why you can preclude other possibilities to define your soft-sediment deformation structures as seismites. Because you didn't explain why the deformation of soft sediments came from seismic activity, not from other reasons.
(5) The structure of the submission can be improved as the author merged all of observations and interpretations together in each section of Chapter 4, which is improper for the readers to think and identify by themselves. Each paragraph is recommended to split into two paragraphs. One is description (with all objective descriptions and no subjective interpretation at all). The second paragraph is interpretations based on descriptions and references.
(6) Font size, legend, scale and direction are improper drawn or missing. Please check the modification and addition carefully.
(7) The sub-member of each sample is marked in Figure 1d, but it is not marked in the picture introduction, so it is recommended to add.
Round 2
Reviewer 1 Report
Dear Authors,
Thank you for submitting the revised version. The manuscript is significantly improved. The manuscript is much more ordered and easy-to-follow now. The changes are valuable.
There are still some issues that I would like to note to improve the manuscript:
11. Some terms are not used consistently in text, figures and tables (e.g. liquefied sand dykes / liquefied sandstone dykes / liquefied stone dykes).
22. I am still not fully convinced that the ‘seismic’ origin of deformation was the only possible mechanism. In my opinion, there are not enough evidence presented, and it seems still to be a hypothesis or assumption.
Detailed comments and suggestions:
Line 16: “liquefied sandstone dyke.
I suggested to use “liquefied sand dyke” in my first review instead, because the rock was not consolidated during the soft deformation, so I think it is more logic to call it sand. You changed this in the tables (Table 1 and partially Table 2), but not in text and figures (Fig. 5). The use of these names is inconsistent across the text and tables. So, it still needs correction to make it consistent in text, all tables and all figures.
Line 36: “ranges” instead of “is range”?
Lines 111-122: This paragraph is very helpful. But, please consider whether you can move it to some other place in the text. Here, in the beginning of ‘Results’ it seems not proper.
Table 1. “Liquefied sand dyke” or “Liquefied sandstone dyke” – it is not consistent with the text, tab. 2 and fig. 5.
Line 142 and everywhere, where you use ‘liquefied sandstone dyke’.
As I mentioned above - – it is not consistent with the tables and fig. 5.
Line 254 and everywhere, where you use ‘microcracks’ and ‘microfractures’ or ‘microfaults’.
I see no response for this comment/suggestion from the first revision, so I repeat. Are those features really in microscale? I can see no results of thin sections analysis etc. In my opinion you should call them ‘small-scale cracks/fractures/faults’.
Table 2. In some places (well B246, X44) you use – “liquefied stone dyke”. It should be “liquefied sand dyke” instead.
Author Response
Dear Editor and Reviewers:
On behalf of my co-authors, thank you for your letter and for the comments concerning our manuscript entitled " Sedimentary Responses of Late Triassic Soft-Sedimentary De-formation to Paleoearthquake Events in the Southwest of North China Plate " (minerals-1823910). We should appreciate the time and effort expend on our manuscript. Those comments are all valuable and very helpful for revising and improving our paper, as well as the important guiding significance to our researches.
We have studied the comments carefully and have made correction that we hope meet with approval. Key revisions were highlighted by red color in the revised manuscript as required.
After carefully read these comments and advices, we conclude the explanations and responses as follows (red part is the comments):
(1) I suggested to use “liquefied sand dyke” in my first review instead, because the rock was not consolidated during the soft deformation, so I think it is more logic to call it sand. You changed this in the tables (Table 1 and partially Table 2), but not in text and figures (Fig. 5). The use of these names is inconsistent across the text and tables. So, it still needs correction to make it consistent in text, all tables and all figures.
Response: Thank you for your comments. In the manuscript, all the words "likefied sandstone dyke" were changed to "likefied sand dyke". Key revisions were highlighted by red color.
(2) Line 36: “ranges” instead of “is range”?
Response: Thank you for your comments. "ranges" replaces "range" in line 36.
(3) Line 254 and everywhere, where you use ‘microcracks’ and ‘microfractures’ or ‘microfaults’. I see no response for this comment/suggestion from the first revision, so I repeat. Are those features really in microscale? I can see no results of thin sections analysis etc. In my opinion you should call them ‘small-scale cracks/fractures/faults’.
Response: Thank you for your comments. We changed all the 'microcracks' and 'microfractures' or 'microfaults' in the manuscript to 'cracks' and 'fractures' or 'faults'.
(4) Lines 111-122: This paragraph is very helpful. But, please consider whether you can move it to some other place in the text. Here, in the beginning of ‘Results’ it seems not proper.
Response: Thank you for your comments. We transferred the contents of lines 111-122 to lines 60-70. Key revisions were highlighted by red color.
(5) I am still not fully convinced that the ‘seismic’ origin of deformation was the only possible mechanism. In my opinion, there are not enough evidence presented, and it seems still to be a hypothesis or assumption.
Response: Thank you for your comments. In the introduction, we describe the meaning of seismites (The original definition of seismites has been extended from the deformation structure caused by seismic events to the deformation structure caused by seismically-induced tsunamis, turbidity currents and other events.).
Moreover, the deformation of soft sediments of the Chang-7 Member occurred in in-situ deposits that were not transported (lines 60-70). Therefore, we believe that the ‘seismic’ origin of deformation was the only possible mechanism.
